# ULoRA: Universal Low-Rank Adaptation of Diverse Deep Learning Architectures

## Abstract

To train Large Language Models (LLMs) having a large number of parameters, the Parameter-Efficient Fine Tuning (PEFT) method based on LoRA, which allows fine-tuning with fewer parameters, is widely employed. However, these methods are primarily designed for application to Transformer architectures, which presents challenges when attempting to apply them to models such as Mamba. To address this limitation, this work proposes Universal LoRA (ULoRA), which applies a Low-Rank Adapter to all deep learning models at the level of universally common blocks. ULoRA achieves generalizability by applying Low-Rank Adapters to blocks, making it applicable to models that do not utilize Transformer architectures. Furthermore, by grouping multiple blocks and applying a single Low-Rank Adapter, ULoRA provides structural flexibility that allows a further reduction in the number of parameters. This significantly reduces resource usage and inference time, making it well-suited for on-device environments with limited resources, while only incurring a slight performance loss. Additionally, if all blocks are grouped to use a single Low-Rank Adapter, task switching during inference is enabled by computing only the adapter. Experimental results show that, for LLaMA-3-8B, ULoRA achieves comparable performance to LoRA with only about 60% of the parameters, while delivering up to 8% higher throughput. For Mamba-2.8B, ULoRA outperforms LoRA with only about 20% of the parameters. In scenarios with limited available resources, ULoRA can be applied using just 4% of the parameters of LoRA, with only a 10% reduction in performance.

## 1 Introduction

Large Language Models (LLMs), which perform tasks such as summarization, translation, prediction, and generation based on knowledge obtained from large datasets, have seen rapid advancements recently in the field of Natural Language Processing (NLP). LLMs such as GPT-3 (175B) (Brown et al. (2020)), PaLM (540B) (Chowdhery et al. (2022)), and LLaMA (70B, 400B) (Touvron et al. (2023a), Touvron et al. (2023b), AI@Meta (2024)) have demonstrated impressive performance across a wide range of tasks. Moreover, models like VisionTransformer (Dosovitskiy et al. (2021)) have achieved high performance in the field of computer vision. However, as the parameter size of these models continues to grow, the economic costs associated with their training and deployment have also increased dramatically.

To address this issue, Parameter-Efficient Fine-Tuning (PEFT) (Mangrulkar et al. (2022)) methods have been proposed. PEFT allows fine-tuning by updating only a portion of the model parameters, thus significantly reducing computational costs while still achieving comparable or equivalent performance. A representative PEFT method is Low-Rank Adaptation (LoRA) (Hu et al. (2021)), which adds Low-Rank Adapters to some parameters of the model and trains only the adapters, without directly training the model itself. This approach can reduce the number of parameters required for training by up to 10,000 times. Moreover, during inference, LoRA merges the adapter weights with the model parameters, thereby eliminating any structural overhead. While LoRA significantly reduces the number of trainable parameters by adding a simple structure, it is not applicable to all models. Since LoRA mainly supports linear layers, it is challenging to apply it to models like Mamba (Gu & Dao (2024)) and ResNet (He et al. (2015)).

To find a way to apply Low-Rank Adapters to all deep learning models, we analyzed the common structural components of these models. Most deep learning models, including LLMs, use an internal structure divided into blocks (layers). In this paper, we define the largest common structural unit within models as the "*Outer Block*" ($\mathcal{BL}_O$) and propose Universal LoRA (ULoRA), which applies Low-Rank Adapters to these $\mathcal{BL}_O$. By dividing models into $\mathcal{BL}_O$ and applying Low-Rank Adapters at this level, ULoRA achieves general applicability across different models. Furthermore, ULoRA offers structural flexibility, as it is also possible to group multiple $\mathcal{BL}_O$ into a larger block and apply a single Low-Rank Adapter. For instance, by grouping two $\mathcal{BL}_O$ together, the number of $\mathcal{BL}_O$, and consequently the number of adapters, is reduced by half. This reduction in the number of adapters allows for the construction of adapters with fewer parameters compared to LoRA.

In resource-constrained environments, such as on-device models for mobile devices, ULoRA provides a significant advantage by allowing extreme reductions in parameter count, leading to considerable decreases in resource usage and inference time. If all $\mathcal{BL}_O$ are grouped into a single Low-Rank Adapter, the last hidden state remains unchanged during inference. Therefore, for task switching, only the replaced adapter needs to be computed, resulting in substantial reductions in inference time.

We proposed in this work, ULoRA, which applies Low-Rank Adapters—traditionally limited to linear layers—to the largest common structure of a model, termed the $\mathcal{BL}_O$, making it applicable to diverse deep learning models. The key contributions of the proposed approach are as follows:

1. **Generality**: ULoRA can be applied to any neural network structure that can be divided into blocks. It is applicable not only to Transformer architectures like LLaMA-3 (AI@Meta (2024)), but also to other architectures such as Mamba (Gu & Dao (2024)) and ResNet (He et al. (2015)), demonstrating its broad applicability.

2. **Structural Flexibility and Efficiency**: ULoRA allows combining multiple $\mathcal{BL}_O$ to apply a single Low-Rank Adapter. This significantly reduces the number of parameters, thereby decreasing resource usage and inference time, providing structural flexibility and efficiency. This makes ULoRA well-suited for on-device models with limited resources.

3. **Ease and Efficiency of Task Switching**: FullStep, which combines all $\mathcal{BL}_O$ into a single Low-Rank Adapter, eliminates the need to compute the model multiple times during inference. If the last hidden state of the model is known, only the new adapter needs to be computed, significantly reducing inference time, thereby facilitating efficient task switching. This is particularly useful in on-device environments.

We performed various experiments to verify the generalizability, structural flexibility, and task-switching ease of ULoRA. First, to evaluate its generalizability, we compare the performance of various adapters on two models, LLaMA-3-8B and Mamba-2.8B. LLaMA-3-8B achieves comparable performance to LoRA while using approximately 60% of the parameters. On Mamba-2.8B, ULoRA achieves higher performance than LoRA with only about 20% of the parameters. Second, to assess the structural flexibility and efficiency, we apply ULoRA to large blocks formed by grouping multiple smaller blocks. In LLaMA-3-8B, ULoRA shows a performance drop of about 10% compared to LoRA but uses only 4% of the parameters, with up to an 8% improvement in throughput. Finally, to confirm the ease of task switching, we compare the inference time of ULoRA and LoRA by applying ULoRA with a single adapter across all blocks.

## 2 RELATED WORK

LoRA (Hu et al. (2021)) is one of the Parameter-Efficient Fine-Tuning (PEFT) methods that addresses computational challenges during the fine-tuning of pre-trained models. LoRA performs low-rank decomposition on the weight matrix $W \in \mathbb{R}^{d \times d}$ by representing it as $W \in \mathbb{R}^{d \times r} \times \mathbb{R}^{r \times d}$, allowing fine-tuning to be conducted using only a small subset of parameters. The model parameters are frozen, and Low-Rank Adapters are attached to some of the parameters, with only the adapters being trained. During inference, the adapter and parameter weights are simply merged, effectively eliminating the adapter structure, and thereby preventing the introduction of any additional overhead.

Since the introduction of LoRA, various modifications and improved architectures have been proposed. VeRA (Kopiczko et al. (2024)) is a vector-based random matrix adaptation method that sig-

nificantly reduces the number of trainable parameters while maintaining performance comparable to LoRA. This method uses a pair of low-rank matrices common across all layers, while instead training a small scaling vector. AdaLoRA (Zhang et al. (2023)) assigns greater parameter budgets to adapters deemed more important in the original LoRA's Low-Rank Adapter, thus making LoRA more adaptive by reducing the parameters allocated to less important weights. DoRA (Liu et al. (2024a)) decomposes the pre-trained weights into two components: magnitude and direction, and applies LoRA to update only the directional component, minimizing the number of trainable parameters during fine-tuning.

However, all of these LoRA variations are primarily focused on application to Transformer architectures. In contrast, Mamba (Gu & Dao (2024)) either cannot utilize LoRA or demonstrates subpar performance when applied. On the other hand, the Universal LoRA (ULoRA) proposed in this paper applies Low-Rank Adapters to the largest common block structure found in deep learning models, enabling it to be used with models that do not utilize Transformer architectures. Additionally, ULoRA allows multiple blocks to be grouped and a single Low-Rank Adapter to be applied across them; if all blocks are grouped into a single Low-Rank Adapter, only the adapter needs to be computed during inference, thereby facilitating efficient task switching.

## 3 METHOD

### 3.1 ULoRA ARCHITECTURE

ULoRA is an improved architecture over LoRA (Hu et al. (2021)), which applies Low-Rank Adapters to the largest common block in a model. Since it is applied to block structures, ULoRA offers general applicability even beyond Transformer architectures. It also provides structural flexibility by allowing multiple blocks to be grouped together and a single Low-Rank Adapter $BA$ to be applied. If all blocks are grouped to use only one Low-Rank Adapter, task switching during inference becomes more efficient since only the adapter computations are needed.

To provide a more detailed explanation, we employ LLaMA-3 (AI@Meta (2024)), which can accommodate both ULoRA and LoRA, as a representative architecture. Fig. 1 shows the application of LoRA and ULoRA to the *Decoder Block* ($\mathcal{BL}_D$) of LLaMA-3-8B. Since both the $\mathcal{BL}_D$ and the *Attention Block* ($\mathcal{BL}_A$) can be considered block structures, which may lead to confusion, we refer to the largest outermost structural unit, such as the $\mathcal{BL}_D$, as the $\mathcal{BL}_O$. Blocks within the $\mathcal{BL}_O$, such as the $\mathcal{BL}_A$, are referred to as *Inner Blocks*. The key difference between ULoRA and LoRA lies in the structure to which the Low-Rank Adapter is applied. As shown in Fig. 1a, Low-Rank Adapters in methods like LoRA, DoRA, and AdaLoRA are typically applied to components such as the Q and V weights in $\mathcal{BL}_A$ in the form of $B_Q A_Q$ and $B_V A_V$. In contrast, ULoRA applies the Low-Rank Adapter to the $\mathcal{BL}_O$, specifically the $\mathcal{BL}_D$, as shown in Fig. 1b.

Because ULoRA is applied at the $\mathcal{BL}_O$ level, it can add Low-Rank Adapters without needing access to the Inner Blocks, making it generally applicable to any deep learning model that can be divided into blocks. Furthermore, as detailed in Section 3.2, ULoRA can also be applied by considering multiple blocks as a single large block.

Since ULoRA and LoRA apply Low-Rank Adapters to different structural levels, their computations within the $\mathcal{BL}_D$ also differ. The Low-Rank Adapter in LoRA is computed within the $\mathcal{BL}_A$, whereas the Low-Rank Adapter in ULoRA is computed outside the $\mathcal{BL}_D$. Therefore, we present two equations to show the computations for both the $\mathcal{BL}_A$ and the $\mathcal{BL}_D$. In the $\mathcal{BL}_A$ as shown in Eq. 1 of LoRA, each Low-Rank Adapter, $B_Q A_Q$ and $B_V A_V$, is computed and added to the target layer. In the $\mathcal{BL}_D$ as shown in Eq. 2, no Low-Rank Adapter is present. Since the Low-Rank Adapter computations are performed within the $\mathcal{BL}_A$, LoRA may face challenges if Transformer architecture is not employed.

$$h_{\mathcal{BL}_A} = softmax\left(\frac{(Q + B_Q A_Q)K^T}{\sqrt{d_k}}(V + B_V A_V)\right) \tag{1}$$

$$h_{\mathcal{BL}_O} = \mathcal{D}(\sigma(\mathcal{G}(h_{\mathcal{BL}_A}) * \mathcal{U}(h_{\mathcal{BL}_A}))) \tag{2}$$

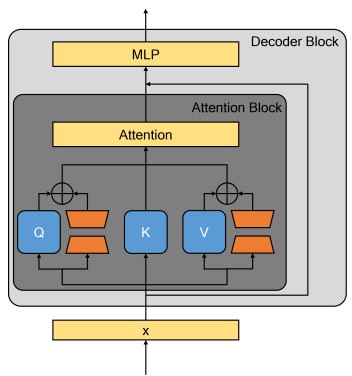 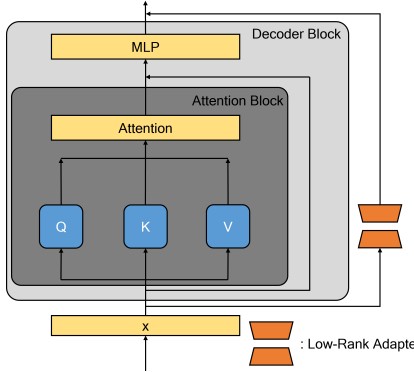

(a) Structure of LoRA applied to Transformer Decoder Block.

(b) Structure of ULoRA applied to Transformer Decoder Block.

Figure 1: Comparison of LoRA and ULoRA architectures.

Where, $\mathcal{D}$, $\mathcal{U}$, and $\mathcal{G}$ are down, up and gate linear layers in Attention-MLP block(Vaswani et al. (2023)), respectively. While, $\sigma$ is the activation function, and $*$ denotes the element-wise multiplication.

ULoRA is applied to the $\mathcal{BL}_O$, specifically the $\mathcal{BL}_D$, and therefore the value of the Low-Rank Adapter is added after all the computations in the $\mathcal{BL}_D$ are completed. In the ULoRA $\mathcal{BL}_A$ as shown in Eq. 3, no Low-Rank Adapter is present, whereas the $\mathcal{BL}_D$ in Eq. 4 includes the Low-Rank Adapter $BA$. Since the computation of the Low-Rank Adapter $BA$ is not needed while calculating the $\mathcal{BL}_O$, ULoRA can be applied to any deep learning model as long as the $\mathcal{BL}_O$ can be defined.

$$h_{\mathcal{BL}_A} = softmax\left(\frac{QK^T}{\sqrt{d_k}}V\right) \tag{3}$$

$$h_{\mathcal{BL}_O} = \mathcal{D}(\sigma(\mathcal{G}(h_{\mathcal{BL}_A}) * \mathcal{U}(h_{\mathcal{BL}_A}))) + BA \tag{4}$$

Since ULoRA is applied to the $\mathcal{BL}_O$, it has the advantage of being applicable to deep learning models that do not use Transformer architectures. Fig. 2 shows the application of ULoRA to ResNet (He et al. (2015)) and Mamba (Gu & Dao (2024)). The ResNet computer vision model is divided into Residual Blocks, which can be considered $\mathcal{BL}_O$ and whose formula is (5). For ResNet-34, where there are 3, 6, 4, and 3 identical Residual Blocks respectively, ULoRA can be applied by considering these as $\mathcal{BL}_O$. Mamba (Gu & Dao (2024)), which is a large language model based on the structured state space model (SSMs), can also be divided into Mamba Blocks. With 64 Mamba Blocks, Mamba-2.8B can apply ULoRA by treating these as $\mathcal{BL}_O$, and the formula is (6). In the ULoRA equation (7), Since ULoRA does not intervene in the internal computations of the $\mathcal{BL}_O$, it has general applicability regardless of the internal computational method.

$$\mathcal{BL}_{O\_ResNet} = \mathcal{F}(x) + x \tag{5}$$

$$\mathcal{BL}_{O\_mamba\_t} = (1 - \sigma(\mathcal{L}(x_t)))h_{t-1} + \sigma(\mathcal{L}(x_t))x_t \tag{6}$$

$$h = \mathcal{BL}_O x + \Delta\mathcal{BL}_O x = \mathcal{BL}_O x + BAx \tag{7}$$

## 3.2 Step with Combined Block

Since Low-Rank Adapters are applied at the block level, the definition of a block can be further extended. It is possible to apply ULoRA by combining multiple $\mathcal{BL}_O$ into a single larger block, thereby further reducing the number of parameters. In this paper, we define the approach of combining multiple $\mathcal{BL}_O$ and applying a single Low-Rank Adapter as a "Step." For example, when applying 2Step ULoRA to LLaMA-3-8B, which has 32 $\mathcal{BL}_D$s, as shown in Fig. 3a, two $\mathcal{BL}_O$ are grouped together, and a Low-Rank Adapter is applied. The number of Low-Rank Adapters is the number of $\mathcal{BL}_O$ divided by the Step, resulting in 16 Low-Rank Adapters for the 2Step example of

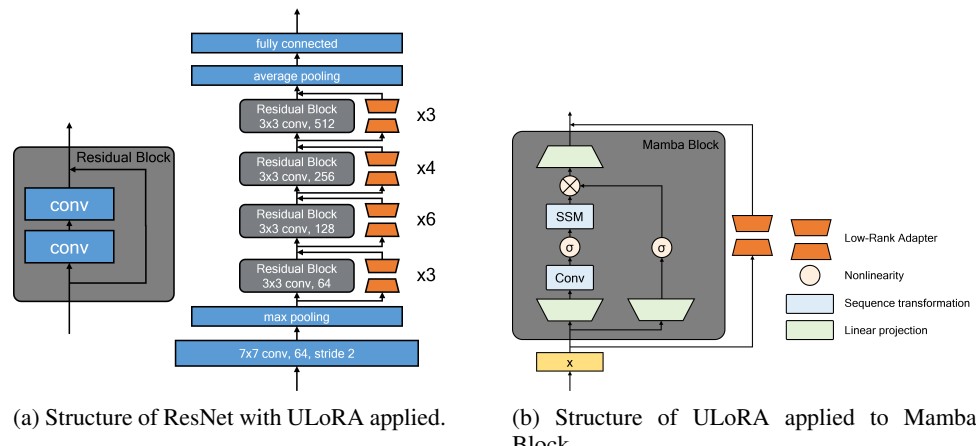

(a) Structure of ResNet with ULoRA applied.

(b) Structure of ULoRA applied to Mamba Block.

Figure 2: Application of ULoRA to non-Transformer architectures.

LLaMA-3-8B. Applying 4Step ULoRA, as depicted in Fig. 3b, results in 8 Low-Rank Adapters. It is also possible to apply ULoRA by grouping all $\mathcal{BL}_O$ into one, which is defined in this paper as "FullStep." When FullStep ULoRA is applied, as shown in Fig. 3c, only a single Low-Rank Adapter is used. The ability of ULoRA to reduce the number of Low-Rank Adapters makes it advantageous for resource-constrained on-device environments.

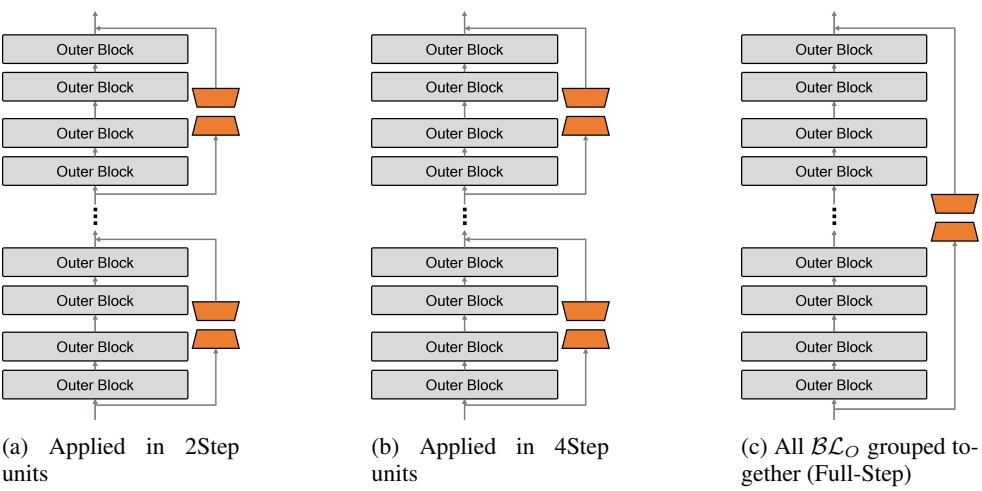

(a) Applied in 2Step units

(b) Applied in 4Step units

(c) All $\mathcal{BL}_O$ grouped together (Full-Step)

Figure 3: Structure of ULoRA with applied Steps.

## 3.3 TASK SWITCHING WITH FULLSTEP

Fig. 4 illustrates an example of task switching using three FullStep ULoRAs. In the forward process of a model using FullStep ULoRA, the adapter is applied after the computation of the last hidden layer is completed. Therefore, if the model's last hidden state is stored during inference, task switching can be achieved by computing only the adapter. For instance, if the value obtained from applying ULoRA1 Adapter is desired after the model's inference, it is sufficient to add the value of ULoRA1 to the model's last hidden state, requiring only the additional computation of ULoRA1. Similarly, ULoRA2 and ULoRA3 can be applied by performing only their respective computations and adding them to the last hidden state. This method is challenging to apply to autoregressive models that predict the next token but can be a powerful method in autoencoder models or vision tasks where different adapters must be used for each task.

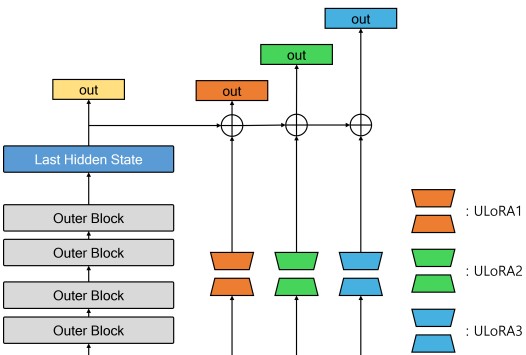

Figure 4: Structure of ULoRA in FullStep.

# 4 EXPERIMENTS AND RESULTS

## 4.1 ENVIRONMENT SETTINGS

The overall experimental environment is described including used models and datasets in this section. Experiments were conducted using the LLaMA-3-8B and Mamba-2.8B models. LLaMA-3 (AI@Meta (2024), Dubey et al. (2024)) is an autoregressive language model by Meta AI, based on the Transformer architecture, and supports a variety of model sizes such as 8B, 70B, and 400B. Mamba (Gu & Dao (2024)) is a novel state space model (SSM) architecture that demonstrates excellent performance on data with high information density, such as language modeling. It is based on a structured state space model (SSMs) and features an efficient hardware-aware design and implementation.

The dataset used includes two types selected from those employed in the experiments of LoRA (Hu et al. (2021)). The E2E Dataset (Novikova et al. (2017)) is a new dataset for training end-to-end, data-driven natural language generation systems in the restaurant domain, which is ten times larger than existing frequently used datasets in this area. Evaluation for E2E was conducted using five metrics: BLEU, NIST, METEOR, ROUGE-L, and CIDEr. Detailed descriptions of the metrics used for evaluating the models trained on these datasets are provided in Appendix 5.

The models were trained using TRL's (von Werra et al. (2020)) SFTTrainer, and the hyperparameters and hardware used for training and inference are summarized in the Table 6 and Table 7 in Appendix 5, respectively. The models were trained for five epochs, and the learning rate was set to 1e-5. The batch size was set to 8, and the warmup steps were set to 500. The optimizer used was AdamW, and the weight decay was set to 0.01. The label smoothing factor was set to 0.1, and the models were compiled using Torch. The seed was set to 42 to ensure reproducibility.

## 4.2 COMPARISON WITH VARIOUS ADAPTERS

This section evaluates the general applicability of ULoRA by measuring performance using various adapters on two models: LLaMA-3-8B and Mamba-2.8B (which does not utilize a Transformer architecture).

The adapters compared with ULoRA includes LoRA (Hu et al. (2021)), DoRA (Liu et al. (2024a)), and AdaLoRA (Zhang et al. (2023)). LoRA is a Low-Rank Adaptation method that significantly reduces the number of trainable parameters for downstream tasks by freezing pre-trained model weights and injecting learnable rank decomposition matrices into each layer of the Transformer architecture. DoRA is a Weight-Decomposed Low-Rank Adaptation that decomposes pre-trained weights into two components: magnitude and direction, and fine-tunes them using LoRA for the directional component, minimizing the number of learnable parameters efficiently. AdaLoRA is an adapter that adaptively allocates parameter budgets across weight matrices. It parameterizes the incremental updates in the form of singular value decomposition, effectively pruning the singular values of less important updates to reduce the parameter budget while avoiding intensive exact SVD computations.

For all adapters being compared, hyperparameters $r = 4$ and $\alpha = 32$ were used, and they were applied only to Q and V in the LoRA case. ULoRA applied the adapter to the Decoder Layer using 1Step, with $r = 4$ and $\alpha = 32$, the same as other adapters. Since Mamba is not composed of a Transformer architecture and does not have an Attention mechanism, adapters were applied to all linear layers within the Mamba Block, excluding the Embedding Layer.

Table 1 presents the results of training various adapters and models using the E2E Dataset on LLaMA-3-8B and Mamba-2.8B. On LLaMA-3, DoRA shows no significant performance improvement compared to LoRA, despite the increase in parameters. ULoRA, in contrast, reduced the parameters to around 60% of LoRA's while showing slight performance gains across all metrics, outperforming DoRA. AdaLoRA demonstrated more than 20% performance improvement compared to LoRA but remains applicable only to Transformer architectures. For Mamba, which does not use Transformer architecture, ULoRA achieved over 20% performance improvement over LoRA with only 20% of the parameters. Compared to DoRA, ULoRA has approximately 17% of the parameters and still provides more than a 20% performance improvement. AdaLoRA also shows lower performance than ULoRA in this case. In summary, ULoRA reduces the number of trainable parameters without a significant performance drop, and its general applicability is advantageous as it can be applied even without Transformer architecture.

Table 1: Performance of various adapters and models trained with the E2E Dataset

| Model | Adapters | Trainable Parameters | BLEU | NIST | MET | ROUGE-L | CIDEr |
|-------|----------|---------------------|------|------|-----|---------|-------|
| LLaMA 3 | LoRA | 1,703,936 | 0.402 | 6.251 | 0.319 | 0.516 | 0.714 |
| | AdaLoRA | 1,704,192 | 0.540 | 7.738 | 0.431 | 0.657 | 1.813 |
| | DoRA | 1,867,776 | 0.420 | 6.240 | 0.422 | 0.589 | 0.755 |
| | ULoRA (Ours) | 1,048,576 | 0.435 | 6.424 | 0.429 | 0.580 | 0.821 |
| Mamba | LoRA | 6,602,752 | 0.244 | 4.440 | 0.304 | 0.419 | 0.093 |
| | AdaLoRA | 6,603,520 | 0.258 | 4.476 | 0.310 | 0.425 | 0.199 |
| | DoRA | 7,434,240 | 0.268 | 4.762 | 0.322 | 0.440 | 0.135 |
| | ULoRA (Ours) | 1,310,720 | 0.394 | 5.916 | 0.398 | 0.541 | 0.254 |

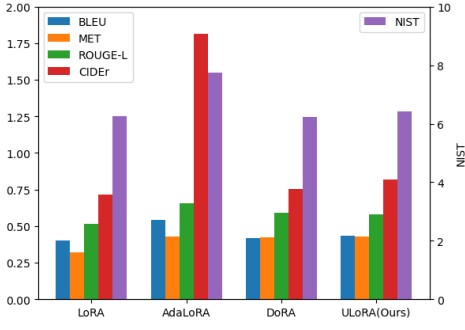 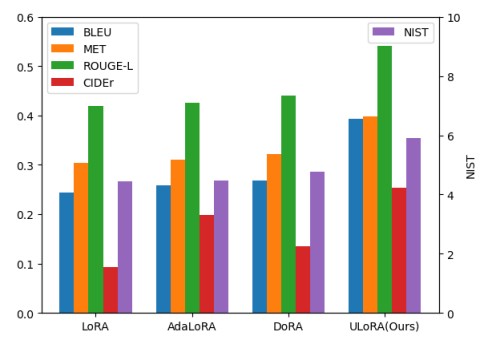

(a) Metrics of adapters applied to LLaMA-3-8B        (b) Metrics of adapters applied to Mamba-2.8B

Figure 5: Metrics graphs of various adapters and models trained with the E2E Dataset. (The secondary axis on the right is for NIST values.)

## 4.3 STEP VARIATION

We evaluate the structural flexibility of ULoRA by training the model with different Step values on the same dataset and measuring the performance. Table 2 shows the performance of ULoRA with different Steps on LLaMA-3-8B trained with the E2E Dataset. $\mathcal{BL}_D$, which is the $\mathcal{BL}_O$ of LLaMA-3-8B, consists of 32 blocks, and the experiment was conducted using six Steps: 1, 2, 4, 8, 16, and 32, and compared against LoRA as shown in Fig. 6a. Most metrics, except for CIDEr, do not show a sharp decline in performance even as the number of parameters decreases. BLEU and NIST follow a similar trend, with ULoRA initially outperforming LoRA at 1Step and gradually

decreasing with increasing Steps, showing a noticeable drop at 32Step. MET consistently shows better performance with ULoRA, approaching LoRA's level at 32Step. ROUGE-L also demonstrates superior performance for ULoRA, becoming similar to LoRA at 4Step. CIDEr drops sharply at 2Step and continues to decrease as Step increases. In summary, ULoRA offers excellent structural flexibility, allowing for various configurations ranging from 1Step to a FullStep that combines all $\mathcal{BL}_O$. The common sharp performance drop at 32Step observed across all metrics suggests that the number of trainable parameters might be too small, prompting additional experiments in Section 4.4.

Table 2: Performance of ULoRA with different Steps

| Adapters | Steps | Number of Adapters | Trainable Parameters | BLEU | NIST | E2E MET | ROUGE-L | CIDEr |
|---|---|---|---|---|---|---|---|---|
| LoRA | | 64 | 1,703,936 | 0.402 | 6.251 | 0.319 | 0.516 | 0.714 |
| ULoRA (Ours) | 1 | 32 | 1,048,576 | 0.435 | 6.424 | 0.429 | 0.580 | 0.821 |
| | 2 | 16 | 524,288 | 0.385 | 5.816 | 0.413 | 0.545 | 0.403 |
| | 4 | 8 | 262,144 | 0.367 | 5.831 | 0.376 | 0.505 | 0.424 |
| | 8 | 4 | 131,072 | 0.380 | 5.803 | 0.393 | 0.525 | 0.331 |
| | 16 | 2 | 65,536 | 0.369 | 5.735 | 0.401 | 0.521 | 0.241 |
| | 32 | 1 | 32,768 | 0.256 | 4.459 | 0.312 | 0.413 | 0.148 |

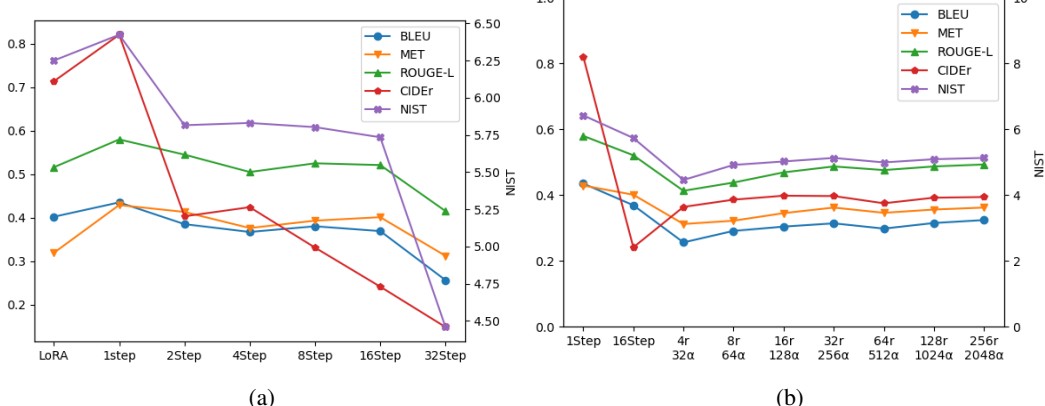

(a)              (b)

Figure 6: Comparison of performance graphs: (a) ULoRA with different Steps, and (b) ULoRA with a fixed Step of 32. (The secondary axis on the right is for NIST values.)

Table 3: Performance with a fixed Step of 32

| Adapters | $r$ | $\alpha$ | Trainable Parameters | BLEU | NIST | E2E MET | ROUGE-L | CIDEr |
|---|---|---|---|---|---|---|---|---|
| ULoRA(Ours)-1Step | 4 | 32 | 1,048,576 | 0.435 | 6.424 | 0.429 | 0.580 | 0.821 |
| ULoRA(Ours)-16Step | 4 | 32 | 65,536 | 0.369 | 5.735 | 0.401 | 0.521 | 0.241 |
| ULoRA(Ours)-32Step | 4 | 32 | 32,768 | 0.256 | 4.459 | 0.312 | 0.413 | 0.364 |
| | 8 | 64 | 65,536 | 0.291 | 4.916 | 0.322 | 0.438 | 0.386 |
| | 16 | 128 | 131,072 | 0.304 | 5.022 | 0.345 | 0.469 | 0.398 |
| | 32 | 256 | 262,144 | 0.314 | 5.129 | 0.362 | 0.487 | 0.397 |
| | 64 | 512 | 524,288 | 0.298 | 4.993 | 0.346 | 0.476 | 0.375 |
| | 128 | 1024 | 1,048,576 | 0.315 | 5.090 | 0.356 | 0.487 | 0.392 |
| | 256 | 2048 | 2,097,152 | 0.324 | 5.130 | 0.362 | 0.493 | 0.394 |

## 4.4 FULLSTEPS

Table 3 and Fig. 6b shows the performance of LLaMA-3-8B trained with the E2E Dataset, using a fixed Step of 32 and varying hyperparameters. We fix the Step value and modify $r$ and $\alpha$ to

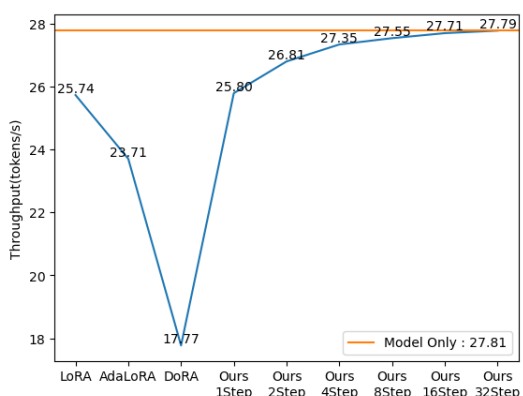

Figure 7: Throughput (tokens/s) by Step for LLaMA-3-8B

measure the performance variation of ULoRA with changes in trainable parameters. Although all results show lower performance compared to 16Step, the performance increases with the number of trainable parameters and starts to converge when $r = 16$. The poor performance observed in the 32Step experiments in Section 4.3 seems to have resulted from an insufficient number of trainable parameters, which prevented proper learning.

## 4.5 THROUGHPUT

We measured the throughput representing the number of tokens processed per unit of time for different adapters. LLaMA-3-8B was used, and the test examples from the E2E Dataset were employed, limiting the maximum sequence length to 1 to predict only a single token. AdaLoRA and DoRA have lower throughput compared to LoRA, whereas our method demonstrates higher throughput, which increases with the Step value. Compared to LoRA, throughput increases by 0.2% at 1Step, 4.1% at 2Step, and up to 8% at 32Step, achieving 99.92% of the throughput of using only the model without any adapters. In summary, ULoRA is one of the LoRA variants that increase throughput, with throughput most closely resembling that of the model without any adapters.

## 4.6 TASKSWITCHING

This section shows the measurement of the speed of Task Switching with FullStep. The LLaMA-3-8B model was used along with test examples from the E2E Dataset. Due to the autoregressive nature of the model, where the output of the forward pass is fed back as input, we limited the maximum sequence length to 1, creating a scenario where only a single token is predicted. Table 4 presents the measured Task Switching speed for LoRA and ULoRA with 32 Steps Adapter inference time is similar for all hyperparameters. Assuming one TaskSwitching, LoRA requires two model forward, which takes $51,084\mu$s. ULoRA requires one model forward and two adapter forward, which does not exceed $22,000\mu$s for all adapters, so the time required for TaskSwitching is overwhelmingly reduced. It takes approximately 185 numbers of TaskSwitchings for ULoRA to catch up to LoRA's TaskSwitching Inference Time of one run. As the number of TaskSwitchings increases, this gap becomes larger and larger, giving ULoRA an advantage in ease and efficiency in TaskSwitching.

## 4.7 FREEZED TRAINABLE PARAMETERS

Fig. 5 presents the performance evaluation when varying the hyperparameters while maintaining a constant number of trainable parameters. When using the same trainable parameters, employing a larger number of low-rank adapters with lower steps yields superior performance. Conversely, when using the same step value, increasing the rank $r$ to augment the trainable parameters generally results in improved performance; however, as illustrated in the results for LoRA and ULoRA-1Step-8r, there are instances where performance declines. Doubling the step value necessitates doubling the parameter size to achieve comparable performance.

Table 4: TaskSwitching Performance on LLaMA-3-8B

| Adapter | $r$ | $\alpha$ | Model Forward | Adapter Forward | Double TaskSwitching | Triple TaskSwitching | number of TaskSwitching |
|---|---|---|---|---|---|---|---|
| LoRA | 4 | 32 | $25{,}902\mu s$ | - | $50184\mu s$ | $75276\mu s$ | $25092 \times n$ |
| ULoRA | 4 | 32 | $20708\mu s$ | $170\mu s$ | $20878\mu s$ | $21048\mu s$ | $20708+170 \times n$ |
| (32Step) | 8 | 64 | | | | | |
| | 16 | 128 | | | | | |
| | 32 | 256 | | | | | |
| | 64 | 512 | | | | | |
| | 128 | 1024 | | | | | |

Table 5: The performance comparison between LoRA and our method for different hyperparameters

| Adapters | Steps | $r$ | $alpha$ | traninable parameters | BLEU | NIST | E2E MET | ROUGE-L | CIDEr |
|---|---|---|---|---|---|---|---|---|---|
| LoRA | | 2 | 16 | 851,968 | 0.420 | 6.188 | 0.426 | 0.583 | 0.568 |
| | | 4 | 32 | 1,703,936 | 0.402 | 6.251 | 0.319 | 0.516 | 0.714 |
| | | 8 | 64 | 3,407,872 | 0.397 | 5.989 | 0.410 | 0.563 | 0.382 |
| ULoRA (Ours) | 1 | 2 | 16 | 524,288 | 0.398 | 5.885 | 0.407 | 0.549 | 0.280 |
| | | 4 | 32 | 1,048,576 | 0.435 | 6.520 | 0.386 | 0.550 | 0.626 |
| | | 8 | 64 | 2,097,152 | 0.408 | 6.032 | 0.414 | 0.553 | 0.383 |
| | 2 | 4 | 32 | 524,288 | 0.396 | 5.994 | 0.412 | 0.552 | 0.486 |
| | | 8 | 64 | 1,048,576 | 0.399 | 6.121 | 0.406 | 0.533 | 0.466 |
| | | 16 | 128 | 2,097,152 | 0.424 | 6.143 | 0.420 | 0.574 | 0.405 |
| | 4 | 8 | 64 | 524,288 | 0.391 | 5.978 | 0.409 | 0.519 | 0.322 |
| | | 16 | 128 | 1,048,576 | 0.381 | 5.870 | 0.396 | 0.522 | 0.338 |
| | | 32 | 256 | 2,097,152 | 0.399 | 5.987 | 0.413 | 0.539 | 0.343 |
| | 8 | 16 | 128 | 524,288 | 0.405 | 5.913 | 0.413 | 0.551 | 0.242 |
| | | 32 | 256 | 1,048,576 | 0.382 | 5.895 | 0.399 | 0.512 | 0.347 |
| | | 64 | 512 | 2,097,152 | 0.401 | 5.910 | 0.410 | 0.549 | 0.301 |
| | 16 | 32 | 256 | 524,288 | 0.390 | 5.897 | 0.405 | 0.535 | 0.263 |
| | | 64 | 512 | 1,048,576 | 0.396 | 5.958 | 0.411 | 0.542 | 0.286 |
| | | 128 | 1024 | 2,097,152 | 0.398 | 5.945 | 0.409 | 0.541 | 0.265 |
| | 32 | 64 | 512 | 524,288 | 0.312 | 5.072 | 0.353 | 0.483 | 0.384 |
| | | 128 | 1024 | 1,048,576 | 0.315 | 5.090 | 0.356 | 0.487 | 0.392 |
| | | 256 | 2048 | 2,097,152 | 0.324 | 5.130 | 0.362 | 0.493 | 0.394 |

## 5 CONCLUSION

We introduced Universal LoRA (ULoRA) in this paper, a comprehensive extension of the Low-Rank Adapter (LoRA) approach to enhance its applicability across diverse deep learning models. Traditional LoRA mainly targets linear layers in Transformer architectures, limiting its scope. ULoRA, however, applies to the largest universally common structural unit—termed the Outer Block—enabling its use in a variety of architectures, including Mamba and ResNet. ULoRA's design allows multiple Outer Blocks to be managed by a single adapter, reducing trainable parameters, resource consumption, and inference time. This makes it particularly effective for resource-limited on-device models. The FullStep configuration, where a single adapter controls all Outer Blocks, enables efficient task switching at inference by simply updating the adapter. Experimental results demonstrated that ULoRA matches or exceeds the performance of methods like LoRA, AdaLoRA, and DoRA with fewer parameters. For LLaMA-3-8B, ULoRA achieved similar performance using 60% of the parameters and provided up to 8% higher throughput. In Mamba-2.8B, ULoRA outperformed LoRA using just 20% of the parameters. Its adaptability was also evident across different Step configurations, maintaining strong performance with fewer parameters. In conclusion, ULoRA expands the applicability of parameter-efficient fine-tuning while reducing resource demands, making it suitable for efficient inference and quick task adaptation. Future work will focus on optimizing ULoRA for autoregressive models and exploring new integrations with emerging deep learning paradigms.

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

APPENDIX: EXPERIMENTAL METRICS

1. **Bilingual Evaluation Understudy Score (BLEU)** (Papineni et al. (2002)) is a method for evaluating the quality of machine translation by comparing the similarity between the machine-generated translation and a reference human translation, based on n-grams.

2. **National Institute of Standards and Technology(NIST)** is a method for evaluating the quality of text which has been translated using machine translation.

3. **Metric for Evaluation of Translation with Explicit ORdering(METEOR)**(Lavie & Agarwal (2007)) is a metric for the evaluation of machine translation output. The metric is based on the harmonic mean of unigram precision and recall, with recall weighted higher than precision. It also has several features that are not found in other metrics, such as stemming and synonymy matching, along with the standard exact word matching.

4. **Recall-Oriented Understudy for Gisting Evaluation-Longest(ROUGE)**(Lin (2004)) is a set of metrics and a software package used for evaluating automatic summarization and machine translation software in natural language processing. ROUGE-L is Longest Common Subsequence(LCS) based statistics. Longest common subsequence problem takes into account sentence-level structure similarity naturally and identifies longest co-occurring in sequence n-grams automatically.

5. **Consensus-based Image Description Evaluation(CIDEr)**(Vedantam et al. (2015)) is for evaluating image descriptions that uses human consensus.

APPENDIX: EXPERIMENTAL PARAMETERS AND CONFIGURATIONS

Table 6: Hyperparameters for Training and Inference

| **Training Parameters** | |
| --- | --- |
| Floating Point | BFloat16 |
| Training Epoch | 5 |
| Learning Rate | 1e-5 |
| Batch Size | 8 |
| Warmup Steps | 500 |
| Optimizer | AdamW |
| Weight Decay | 0.01 |
| Label Smoothing Factor | 0.1 |
| Torch Compile | True |
| Seed | 42 |
| **Inference Parameters** | |
| Beam Search | 10 |
| Max New Tokens | 50 |
| No Repeat Ngram Size | 4 |
| Length Penalty (E2E) | 0.9 |
| Length Penalty (DART) | 0.8 |

Table 7: Hardware for Training and Inference

| | Training | Inference |
| --- | --- | --- |
| CPU | i9-13900K | Ryzen 5700X |
| RAM | 128GB | 128GB |
| VGA | RTX 4090 24GB $\times$ 2 | RTX 3090 24GB $\times$ 1 |

