# OpenReview forum: "ULoRA: Universal Low-Rank Adaptation of Diverse Deep Learning Architectures"
_ICLR.cc/2025/Conference — ICLR 2025 Conference Withdrawn Submission_

### Official Review · Reviewer_QHrg · 2024-10-30

**Soundness:** 2
**Presentation:** 2
**Contribution:** 2
**Rating:** 3
**Confidence:** 3

**Summary:**

This paper introduces Universal LoRA (ULoRA), a Parameter-Efficient Fine Tuning (PEFT) method that extends Low-Rank Adapters (LoRA) to models beyond Transformers, such as Mamba. ULoRA operates by applying a Low-Rank Adapter at the level of universally common blocks within the model, enhancing generalizability across various architectures. By grouping multiple blocks and using a single adapter, ULoRA reduces the total number of parameters involved in fine-tuning, which significantly improves memory and computational efficiency. The method proves especially suitable for on-device environments with limited resources, enabling efficient task-switching during inference. Experimental results show that ULoRA can achieve comparable or superior performance to traditional LoRA while utilizing fewer parameters and maintaining higher throughput.

**Strengths:**

**Broad Applicability**: ULoRA extends PEFT beyond Transformer-based models, making it applicable to a wider range of architectures.

**Flexible Structure**: Grouping blocks and using a single Low-Rank Adapter allows efficient task-switching during inference, enhancing adaptability in real-time applications.

**Weaknesses:**

There is a lack of experimental comparisons and discussion; please see the questions for more details. ,

Some figures and tables in the experimental section are somewhat redundant. For example:
1. Table 1 conveys the same information as Figure 5.
2. Similarly, Table 2 and Table 3 are highly repetitive.
3. Figure 5 is not correctly referenced in the text.

Overall, I think the experiment section needs to be reorganized.

**Questions:**

1. LoRA can merge adapter weights with model parameters, thereby eliminating structural overhead during inference. However, the proposed ULoRA does not appear to be mergeable with the original model during inference. If this is correct, how does ULoRA achieve higher throughput compared to LoRA (as shown in Fig. 7)?

2. Compared to Llama, ULoRA achieves significantly higher task performance on Mamba over LoRA. Thus, it would be helpful to see throughput comparison results on Mamba as well.

3. In addition to throughput, memory consumption is another important metric for efficient fine-tuning. I think providing memory consumption comparisons for both training and inference would help to further validate the effectiveness of your approach.

---

### Official Review · Reviewer_Ppng · 2024-11-02

**Soundness:** 3
**Presentation:** 3
**Contribution:** 3
**Rating:** 5
**Confidence:** 2

**Summary:**

This paper proposes a new variant of LoRA, called ULoRA(U means universal). The original LoRA only adapts new trainable parameters to linear modules, while ULoRA applies them at the level of universally common blocks. The paper conducts various experiments on different model structures like transformers and SSMs, showing that ULoRA has less trainable parameters, higher throughput, less task switch time, while maintain the accuracy.

**Strengths:**

- This paper propose a solution for LoRA adaption of different model structures that previous works have not been given much attention like CNN and SSM.
- This article focuses not only on the method's advantages in terms of algorithmic accuracy but also on topics such as throughput and task switching speed, which are closer to real application scenarios.

**Weaknesses:**

- "Since Low-Rank Adapters are applied at the block level, the definition of a block can be further extended." in line 210. I don't really understand the author's motivation for doing this. Because it would cause the adapter to not be able to merge with the main weight in the reasoning phase, destroying the original design idea of LoRA.
- In line 154, you said "Since the Low-Rank Adapter computations are performed within the BLA, LoRA may face challenges if Transformer architecture is not employed", you use this to assert that it is impossible to use LoRA in model architectures like CNN, but in fact the official library of [peft](https://github.com/microsoft/LoRA/blob/main/loralib/layers.py) already implements LoRA on convolutional layers, and the vast majority of models nowadays are based on linear layers or variants (like convolution).This so-called ‘limited generality’ of LoRA is doubtful in my opinion.
- In Figure 2(a), you explained how to use ULoRA on CNN based models like ResNet, but in experiment you only evaluate transformers and ssms, It would be nice if the article could be supplemented with experiments on ResNet-like models.

**Questions:**

- An important reason why LoRA can be the most popular parameter efficient fine-tuning method is that its structure is very elegant - i.e., its adapter can be fused with the MAIN weight in the inference phase. While the experiments in the paper demonstrate that ULoRA introduces negligible overhead, it is clear that this cross-layer connection clearly cannot be fused with the backbone. May I ask the authors what they think about this design problem?
- Can you evaluate the TaskSwitching Performance & Throughput on Mamba-like models?

---

### Official Review · Reviewer_Wwr5 · 2024-11-02

**Soundness:** 1
**Presentation:** 2
**Contribution:** 1
**Rating:** 3
**Confidence:** 4

**Summary:**

The paper introduces Universal Low-Rank Adaptation (ULoRA), a parameter-efficient fine-tuning (PEFT) approach that broadens the scope of LoRA’s low-rank adaptation technique to work well beyond Transformer-based architectures. While traditional LoRA methods apply low-rank, trainable matrices specifically to the attention weights in Transformer layers—allowing efficient fine-tuning without altering the primary model parameters—ULoRA extends this method to other deep learning architectures, including Mamba and ResNet. ULoRA’s approach involves injecting low-rank matrices across different layers or grouped blocks, further reducing the number of tunable parameters and adding structural flexibility. This flexibility not only reduces resource usage and speeds up inference but also enables efficient task-switching, as the adapter component alone can be recomputed for new tasks. Although ULoRA is less effective than LoRA variants on Transformer architectures, it demonstrates superior performance on Mamba and ResNet, as evidenced by results on an end-to-end (E2E) dataset. The authors conducted several experiments to test ULoRA’s versatility, structural adaptability, and ease of task switching.

**Strengths:**

- The paper investigates the use of parameter-efficient fine-tuning in universal architectures which is not very common as most studies focus on transformer-based architectures.

- By grouping multiple blocks and applying a single Low-Rank Adapter, ULoRA provides structural flexibility that allows a further reduction in the number of parameters.

**Weaknesses:**

- Novelty Concern: The idea lacks novelty, as it modifies the LoRA technique to apply low-rank adaptation across one or multiple layers/blocks rather than solely on attention weights within layers.

- Soundness of Results: The soundness of the results presented is very shaky. I believe the experiments have been run with very non-optimal settings. For example, in Table 1, the authors get a BLEU score of approximately 0.4 using LoRA and LLaMA-3-8B. It is very common to achieve BLEU scores of approximately 0.7 using models as small as GPT-2 (125 M) parameters.

- Limited Dataset Evaluation: The study's generalizability may be limited, as ULoRA’s performance is validated primarily on a single end-to-end (E2E) dataset. Broader testing across diverse datasets could strengthen the claims of universal applicability.

- Missing Architecture Details and Challenges: Challenges related to applying LoRA on Mamba and ResNet models have not been discussed at all. Although the authors provide performance metrics for the LoRA-based approach in their comparisons, they have not addressed the corresponding implementation framework.

- Missing Baseline and Discussion: A highly relevant paper applies a variant of LoRA across layers for Mamba-based models; the authors should compare qualitatively and quantitatively.
Reference: Low-Rank Interconnected Adaptation across Layers (https://arxiv.org/abs/2407.09946)

**Questions:**

The paper needs significant improvements before being ready to be published - please refer to weaknesses for a detailed analysis. Some additional questions are as follows:

- What extra information does Figure 5 provide in contrast to Table 1? Do they imply the same with basically different ways of presentations? Similar situation with Table 2 and Figure 6.
- Are there guidelines for selecting these hyperparameters based on model size, dataset complexity, or resource limitations?

---

### Note · Authors · 2024-11-25

**Comment:**

We appreciate the comments of reviewers. We would like to withdraw the paper. We feel the paper is not appropriate with the ICLR 2025.

**Withdrawal Confirmation:**

I have read and agree with the venue's withdrawal policy on behalf of myself and my co-authors.